# The First Exploratory Personalized Medicine Approach to Improve Bariatric Surgery Outcomes Utilizing Psychosocial and Genetic Risk Assessments: Encouraging Clinical Research

**DOI:** 10.3390/jpm13071164

**Published:** 2023-07-20

**Authors:** Panayotis K. Thanos, Colin Hanna, Abrianna Mihalkovic, Aaron B. Hoffman, Alan R. Posner, John Busch, Caroline Smith, Rajendra D. Badgaiyan, Kenneth Blum, David Baron, Lucy D. Mastrandrea, Teresa Quattrin

**Affiliations:** 1Behavioral Neuropharmacology and Neuroimaging Laboratory on Addictions (BNNLA), Clinical Research Institute on Addictions, Department of Pharmacology and Toxicology, Jacobs School of Medicine and Biomedical Sciences, University at Buffalo, Buffalo, NY 14203, USA; cshanna2@buffalo.edu (C.H.); amihalko@buffalo.edu (A.M.); 2Department of Psychology, University at Buffalo, Buffalo, NY 14203, USA; 3Department of Surgery, Methodist Hospital Medical Center, Dallas, TX 75001, USA; abh2@buffalo.edu (A.B.H.); ldm@buffalo.edu (L.D.M.); 4Department of Surgery, Jacobs School of Medicine and Biomedical Sciences, University at Buffalo, Buffalo, NY 14203, USA; arposner@buffalo.edu (A.R.P.); jbusch@buffalo.edu (J.B.); 5UBMD Pediatrics, JR Oishei Children’s Hospital, University at Buffalo, Buffalo, NY 14203, USA; cesmith5@buffalo.edu; 6Department of Psychiatry, South Texas Veteran Health Care System, Audie L. Murphy Memorial VA Hospital, Long School of Medicine, University of Texas Health Science Center, San Antonio, TX 78229, USA; badgaiyan@gmail.com; 7Division of Nutrigenomics, SpliceGen, Therapeutics, Inc., Austin, TX 78701, USA; drd2gene@gmail.com; 8Department of Psychiatry, Wright State University Boonshoft School of Medicine, Dayton, OH 45435, USA; dbaron@westernu.edu; 9Division of Addiction Research & Education, Center for Exercise Sports & Global Mental Health, Western University Health Sciences, Pomona, CA 91766, USA; 10The Kenneth Blum Behavioral & Neurogenetic Institute, LLC., Austin, TX 78701, USA; 11Institute of Psychology, ELTE Eötvös Loránd University, 23-27, 1075 Budapest, Hungary; 12Centre for Genomics and Applied Gene Technology, Institute of Integrative Omics and Applied Biotechnology (IIOAB), Nonakuri, Purba Medinipur 721172, West Bengal, India; 13Department of Molecular Biology, Adelson School of Medicine, Ariel University, Ariel 40700, Israel

**Keywords:** obesity, bariatric surgery, Reward Deficiency Syndrome, addiction, psychosocial risk factors, genetic risk assessment

## Abstract

It is predicted that by 2030, globally, an estimated 2.16 billion adults will be overweight, and 1.12 billion will be obese. This study examined genetic data regarding Reward Deficiency Syndrome (RDS) to evaluate their usefulness in counselling patients undergoing bariatric surgery and gathered preliminary data on the potential use in predicting short term (6-month) weight loss outcomes. **Methods**: Patients undergoing bariatric surgery (*n* = 34) were examined for Genetic Addiction Risk Severity (GARS) [measures the presence of risk alleles associated with RDS]; as well as their psychosocial traits (questionnaires). BMI changes and sociodemographic data were abstracted from Electronic Health Records. **Results**: Subjects showed ∆BMI (M = 10.0 ± 1.05 kg/m^2^) and a mean % excess weight loss (56 ± 13.8%). In addition, 76% of subjects had GARS scores above seven. The homozygote risk alleles for *MAO* (rs768062321) and *DRD1* (rs4532) showed a 38% and 47% prevalence among the subjects. Of the 11 risk alleles identified by GARS, the *DRD4* risk allele (rs1800955), was significantly correlated with change in weight and BMI six months post-surgery. We identified correlations with individual risk alleles and psychosocial trait scores. The *COMT* risk allele (rs4680) showed a negative correlation with EEI scores (*r* = −0.4983, *p* < 0.05) and PSQI scores (*r* = −0.5482, *p* < 0.05). The *GABRB3* risk allele (rs764926719) correlated positively with EEI (*r* = 0.6161, *p* < 0.01) and FCQ scores (*r* = 0.6373, *p* < 0.01). The *OPRM1* risk allele showed a positive correlation with the DERS score (*r* = 0.5228, *p* < 0.05). We also identified correlations between DERS and BMI change (*r* = 0.61; *p* < 0.01). **Conclusions**: These data support the potential benefit of a personalized medicinal approach inclusive of genetic testing and psychosocial trait questionnaires when counselling patients with obesity considering bariatric surgery. Future research will explore epigenetic factors that contribute to outcomes of bariatric surgery.

## 1. Introduction

Overweight and obesity were estimated to afflict nearly 1.5 billion adults worldwide in 2008 [1]. By 2030, the respective number of overweight and obese adults was projected to be 1.35 billion and 573 million individuals without adjusting for secular trends. If recent secular trends continue unabated, the absolute numbers were projected to total 2.16 billion overweight and 1.12 billion obese individuals [2]. It is noteworthy that the unwanted burden is greater for much of Asia, Latin America, the Middle East, and Africa due to differences in fat patterning and body composition and the cardiometabolic effects of body mass index (BMI) at levels far below standard BMI overweight cutoffs of 25.

Candidates for metabolic/bariatric surgery show a high prevalence of food addiction (FA). A random-effects meta-analysis was performed with cross-sectional studies to calculate the weighted prevalence of FA at the pre- and postoperative moments. For longitudinal studies, which measured FA at both time points for the same individuals, absolute prevalence reduction (APR) was calculated. Of the 6626 records, 40 studies were included in the meta-analysis. The properative weighted prevalence of FA was 32% (95% CI: 27–37%; 33 groups), whereas the postoperative prevalence was 15% (95% CI: 12–18%; 14 groups). Observational data suggest an attenuation in the prevalence of FA among patients that undergo bariatric surgery [3].

Bariatric surgery is the most efficient weight loss treatment for patients unable to achieve or sustain weight loss using non-surgical strategies [4]. A 2020 study reported weight loss of 47% after laparoscopic sleeve gastrectomy (LSG) and 55% after Roux-en-Y gastric bypass (RYGB) [5]. However, in patients undergoing bariatric surgery, there can be recidivism and unintended negative behavioral consequences [6]. For instance, rates of alcohol use disorder (AUD) post-surgery are as high as 28.4% compared to patients’ AUD rate of 4.5% preoperatively [7].

Reward Deficiency Syndrome (RDS) is a framework for examining the role of genetics and epigenetics related to addictive behaviors [8,9]. This syndrome encompasses impulsive and compulsive behaviors, such as gambling, binge eating, and alcohol and drug abuse, resulting from hypodopaminergic reward circuitry functioning [10]. The Genetic Addiction Risk Severity (GARS) test was developed as a tool to identify risk variations of particular genes (See Appendix A): [dopamine receptor-D1 (*DRD1*), dopamine receptor-D4 (*DRD4*), dopamine transporter (*DAT1*), serotonin transporter gene (*5-HTT*)-linked, monoamine oxidase A (*MAOA*), opioid receptor Mu 1 (*OPRM1*), catechol-o-methyltransferase (*COMT*), and gamma-aminobutyric acid receptor subunit beta-3 (*GABRB3*)] that predict vulnerability to pain [11], addiction, and other compulsive behaviors that fall under the umbrella of RDS [12,13,14,15,16]. Understanding an individual’s GARS score could provide proactive management for individuals at risk for RDS [12].

Patients who are candidates for bariatric surgery often struggle with body image, suboptimal quality of life, and symptoms of depression and anxiety [17]. Prior to surgery, psychological assessments to identify these concerns can guide patients to support/interventions to maximize the likelihood of success following bariatric surgery [4,18]. In spite of these proactive interventions, there is variability in bariatric surgery short and long-term outcomes which may be attributed to AUD and other addictions [19]. This observation highlights the importance of identifying individuals who will need more aggressive preparation pre-surgery and post-operative behavioral follow-up [19,20].

The goal of the present study was to utilize a personalized medicine approach for bariatric surgery patients by examining potential genetic and psychosocial markers that may predict which patients undergoing bariatric surgery are at greater risk for recidivism of obesity and negative outcomes.

## 2. Methods

### 2.1. Participants

Seventy patients were approached during the last pre-surgery consultation appointment at the Kaleida Health Bariatric Center (Buffalo, NY, USA). After describing the study and providing an informational brochure, 34 subjects provided informed consent. The study was approved by the University at Buffalo’s Institutional Review Board. Due to the COVID-19 pandemic, there was an unanticipated lack of in-person follow-up available for patients after surgery resulting in a smaller than anticipated sample size. As a result, sample sizes are not consistent between patients who completed the questionnaire, patients who provided a GARS sample, and patients who could be reached for follow-up appointments at post-op study time points.

### 2.2. Data Collection

Participants’ data collected in the questionnaire were: age, sex, relationship status, education, and income. Additional demographic data and health parameters pre- and post (1, 3, 6 months) bariatric surgery were collected from the electronic health record. Weight loss (kg), change in BMI from preoperative to 6 months post-surgery and percent excess weight loss (%EWL) were calculated. The latter is a standard way for surgeons to assess post bariatric surgery weight changes. The formula for %EWL is: [(initial weight-ost-op weight)/(initial weight-ideal weight)] × 100, where ideal weight corresponded to weight associated with 25 kg/m^2^ BMI [21].

### 2.3. Psychosocial Questionnaires

Participants were given tablets to complete psychosocial questionnaires privately at the time of the visit or given the option to complete via email/web link remotely. Paper format was also available for those who were uncomfortable using technology. The questionnaire administered was a compilation of several validated scales (see Appendix A) used to evaluate psychosocial well-being in domains that are relevant to patients with obesity, such as measures of: nutritional habits: Eating Attitudes Test-26 (EAT-26) [22]; Food Cravings Questionnaire-Trait Reduced (FCQ-TR) [23]; Eating Expectancies Inventory (EEI) [24]; food addiction: modified Yale Food Addiction Scale 2.0 (mYFAS 2.0) [25]; binge-eating disorder symptoms: Weight Influenced Self-Esteem Questionnaire (WISE-Q) [26]; depression and anxiety: Difficulties in Emotion Regulation Scale (DERS) [27]; Center for Epidemiologic Studies Depression Scale (CESDS) [28], and chronic stress and life quality: Chronic Stress Index (CSI) [29,30], sleep: Pittsburgh Sleep Quality Index (PSQI) [31]. Questionnaire data were scored according to the corresponding scoring key included for each scale.

### 2.4. Genetic Addiction Risk Severity (GARS)

The GARS test assessed eleven specific gene polymorphisms known to predict increased susceptibility for substance use disorder (SUD). A cheek swab sample was collected from (*n* = 34) subjects and processed as previously described [32]. Briefly, DNA from the sample was isolated by PCR amplification and assayed for polymorphisms in the following genes: *DRD1*, *OPRM1*, *DRD2*, *DRD3*, *DRD4*, *COMT*, *DAT1*, *DRD4-R*, *GABRB3*, *HTTLPR*, and *MAOA* [14,33]. DNA assay results were analyzed by Geneus Health, (San Antonio, TX, USA) for specific polymorphism markers that have been previously established [33] to predict SUDs as well as RDS. As previously described [13], a risk score for each subject was then calculated based on the genetic analysis.

### 2.5. Statistical Analysis

Data are expressed as mean ± SD and analyzed and graphed using GraphPad Prism software 8.1.2 (San Diego, CA, USA). Linear regression of correlations between outcome variables were analyzed for ∆BMI and ∆Weight at 3- and 6-months post-surgery as well as for each GARS risk allele tested and psychosocial questionnaire results. Post hoc analysis (Tukey’s HSD test, Sidak’s test) was performed for all significant ANOVA outcomes. The change in weight/BMI for each patient was calculated from the bariatric clinic reports at all study time points. These results were included in a correlational analysis with the GARS scores of all patients who provided a buccal swab. The 3- and 6-month time points were included in the statistical analysis to determine correlation between weight/BMI change and overall GARS score. Correlational analyses were also performed for each of the individual risk alleles implicated by GARS with patients’ change in weight/BMI.

### 2.6. Ethics

This study was carried out in accordance with and approval by The Institutional Review Board of the University at Buffalo. All subjects were informed about the study, and all provided signed informed consent. The study was conducted in accordance with the Declaration of Helsinki, and the protocol was approved by the Ethics Committee of HRP-503.

## 3. Results

### 3.1. Baseline Demographic Characteristics

Participants (*n* = 34) were recruited from the Bariatric Program at Kaleida Health, which is designated as a Comprehensive Center under the Metabolic and Bariatric Surgery Accreditation and Quality Improvement Program. This study was approved by the IRB at the University at Buffalo. Table 1 summarizes self-reported demographic data via questionnaires and pre-surgery BMI and weight extracted from electronic health records for 26/34 participants. Participants were predominantly female and Caucasian, with >50% reporting a childhood history of overweight/obesity. Table 2 reports the changes in BMI for 24 subjects after undergoing bariatric surgery. Of the subjects, 74% underwent vertical sleeve gastrectomy. The COVID-19 epidemic prevented us from obtaining psychosocial questionnaires and follow-up data in 10 participants.

### 3.2. Psychosocial and GARS Data

The psychosocial questionnaires scores are summarized in Table 3. Background information on these scores and their significance are provided in Appendix A. A majority of patients reported sleep quality, food cravings, depressive-like symptoms, and food addiction scores similar to those reported in previous studies on obese patients [23,25,28,31]. However, the mean score from the modified Yale Food Addiction Scale (mYFAS) was 2.0 which is lower than anticipated, suggesting a less than expected prevalence of food addiction for this cohort. Additionally, the symptom count in our study (1.32 ± 1.23) was lower than previously reported in other studies on individuals with obesity [34,35].

The GARS test results identified copy number of the 11 risk alleles for each patient (*n* = 34), assigned as either homozygote (two copies of the risk allele), heterozygote (one copy of the risk allele), or low risk (no copies of the risk allele). Figure 1 presents the % frequency for each risk allele for each gene of interest. The *MAO* and *DRD1* genes yielded the greatest frequency of homozygote alleles, whereas none of the subjects were homozygous for risk allele genes *OPRMI*, *DRD4* (rs761010487), and *DAT1f*. A total GARS score of seven or greater qualifies as high risk for alcohol addiction. Our results showed that 76% of subjects fell into the high-risk category for alcohol addiction (Figure 2).

### 3.3. Risk Alleles and Association with Weight Loss and Inventory Scores

We analyzed body weight data available preoperatively and at six months post-surgery (*n* = 21) for the 11 risk alleles analyzed. The *DRD4* risk allele (rs1800955) was significantly correlated with preoperative weight (*r* = 0.4331, *p* < 0.05 Figure 3a), change in weight (*r* = 0.4726, *p* < 0.05; Figure 3b), and change in BMI (*r* = 0.4577, *p* < 0.05; Figure 3c).

Correlations were examined between the various psychosocial questionnaire scores and individual risk alleles. Results showed a significant negative correlation between the *COMT* risk allele and EEI scores (*r* = −0.4983, *p* < 0.05; Figure 3d). The *COMT* risk allele was also negatively correlated with the PSQI scores, (*r* = −0.5482, *p* < 0.05; Figure 3e). In addition, there was a positive correlation between the *GABRB3* risk allele and EEI scores (*r* = 0.6161, *p* < 0.01; Figure 3f) and a positive correlation with scores on the FCQ (*r* = 0.6373, *p* < 0.01; Figure 3g). Next, the *OPRM1* risk allele was positively correlated with the DERS scores (*r* = 0.5228, *p* < 0.05; Figure 3h). Results also showed that the scores from the DERS were also significantly correlated with BMI before surgery (*r* = −0.5142, *p* < 0.01; Figure 3i) and at 6 months post-surgery (*r* = −0.6137, *p* < 0.01; Figure 3j).

### 3.4. Heterosis

To determine if heterosis was occurring within the genes evaluated, a student’s *t* test was conducted to compare data from patients with no risk allele copies for each gene and patients who were heterozygous for the risk allele of each gene. Heterosis is the adaptive superiority of the heterozygous genotype with respect to one or more characters in comparison with the corresponding homozygote [36]. Moreover, in proposing the term *heterosis* to replace the older term *heterozygosis*, the newer idea aimed to avoid limiting the term to the effects that can be explained by heterozygosity in Mendelian inheritance. Specifically, Heterosis is considered to be an outbreeding enhancement which displays an improved or increased function of any biological quality in a hybrid offspring [37]. An offspring is heterotic if its traits are enhanced as a result of mixing the genetic contributions of its parents. Currently, the actual mechanism for this enhanced improvement or effect is unknown.

The analysis revealed a significant difference in Δ in weight (kg) (*t* = 2.400, *p* = 0.03), Δ in BMI (*t* = 2.234, *p* = 0.04), and %EWL (*t* = 2.418, *p* = 0.03) at six months post-surgery between patients with 0 or 1 copies of the *DRD4* risk allele (Figure 4). Subjects with one copy of the *DRD4* risk allele achieved greater weight loss at the six months post-surgery timepoint, including Δ in BMI and %EWL, than subjects with no copy of this risk allele.

## 4. Discussion

In this study of obese adults undergoing bariatric surgery, we obtained psychosocial inventory scores similar to obese subjects in previous studies [38,39,40,41]. As expected, subjects reported greater eating disorder pathology, sleep disturbances, and food cravings as well as lower life enjoyment and satisfaction than historical non-obese controls [42,43,44,45]. There were some inventory score results, however, that were not similar to previous reports in obese subjects. Murray et al. reported that the reduction in food addiction observed six months post-surgery was not maintained at 12 months [46]. In the study by Murray et al., food addiction at 12 months post-surgery was associated with poorer weight loss, eating, and lifestyle behaviors. It is of interest that between baseline and 24 months, YFAS scores decreased (*p* = 0.006) and alcohol intake increased in the surgery group (*p* = 0.005). Significant changes were not observed in the diet or no treatment groups [46].

The genetic results (GARS test) examined the genetic vulnerability for addictive behaviors and RDS as measured by the presence of ten reward genes and eleven risk polymorphisms [13]. The evidence presented provided relevant genetic information that will reinforce targeted therapies to improve recovery and prevent recidivism on an individualized basis, especially in terms of bariatric surgery. The primary driver of RDS is a hypodopaminergic trait (genes) as well as epigenetic states (methylation and deacetylation on chromatin structure). We hypothesize that genetic testing at an early age may be an important strategy to reduce or eliminate pathological substance and behavioral seeking activity post bariatric surgery.

Analysis of the GARS results revealed a diverse genetic profile across study participants. At least one patient from the cohort was homozygous for each of the risk alleles of the genes tested except for the *DRD4* (rs1800955) and *OPRM1* genes, as well as the *DAT1* gene, for which no patient carried the risk allele. The most common homozygous risk alleles present in the study cohort were for the *DRD1* and *MAOA* gene, with a prevalence of 47 and 38 percent, respectively. Using this information, alcohol addiction risk scores were calculated and revealed that 76% of the subjects had an elevated risk for AUD. This outcome highlights the importance of utilizing a personalized approach prior to surgery in order to identify patients that might need additional counselling or intervention after surgery.

The correlational analyses revealed several correlations between GARS and weight changes after bariatric surgery. First, the *DRD4* risk allele (rs1800955) genotype was found to be significantly associated with both BMI and weight change at six months post-surgery. Patients with two copies of the risk allele showed greater change in weight between preop and six months than did patients with no copies of the *DRD4* risk allele. The *DRD4* gene has previously been linked to Attention Deficit Hyperactivity Disorder (ADHD) [47,48]. The D4 receptors in the brain are prominently expressed in the prefrontal cortex, an area that regulates attention, decision-making, and inhibitory control [49]. These receptors are also located in the hippocampus, amygdala, and hypothalamus, where they are involved in the reinforcing properties of food [50]. Each one of these cognitive functions could influence patients’ success at dieting and maintaining weight loss after surgery, mediating the positive relationship we identified between greater change in weight and more copies of the *DRD4* risk allele. Our findings indicate that having two copies of the *DRD4* risk allele, as opposed to one or none, could serve as a beneficial genetic predictor for weight loss after surgery. It is possible that the hypodopaminergic functioning associated with this *DRD4* gene polymorphism lowers the reward experienced from food consumption, giving patients an upper hand when it comes to avoiding food and losing weight.

Correlations of each individual GARS gene and psychosocial inventory outcomes also yielded significant outcomes. For instance, we found that scores on the EEI were correlated negatively with the *COMT* gene and positively with the *GABRB3* gene. The inverse relationship seen between EEI scores and copies of the COMT risk allele, the Val158Met/rs4680 polymorphism, suggests that patients carrying one or more copies expect less mood and cognitive benefits from eating compared to patients with no copies of this allele who scored higher on the inventory. An important regulator of dopaminergic neurotransmission, *COMT* is a degrading enzyme and metabolizes DA to inactive compounds, especially in prefrontal brain regions [51,52]. Prior studies have uncovered a potential connection between *COMT* polymorphisms and eating disorder pathology. For instance, an investigation into the influence of the *COMT* polymorphism on behavioral impulsivity in the binge eating disorder found that patients within the BED group that were homozygous for the risk allele showed stronger deficits in inhibitory control [53]. Given the current literature on *COMT* and eating behavior, we expected to see higher scores of eating expectancies in patients with one or more copies of the risk allele. Future data at 12 months post-surgery may reveal more insight into the correlation identified here or present a possible shift in the direction of this relationship.

The significant correlation between EEI scores and the *GABRB3* gene indicates that patients with a homozygous risk allele genotype, the 181 variants of the gene, reported greater expectancies of food to improve their cognitive and emotional states. Disruption in the expression of GABA_A_ receptors is associated with imbalances in excitatory/inhibitory signaling in the brain that can have profound consequences on cognition, perception, and maintenance of mood [54]. Dysregulation brought on by the *GABRB3* polymorphism involves an increase in GABA transmission that leads to a reduction in DA release at the nucleus accumbens. The EEI discussed here contains items related specifically to affective mood regulation. A major factor influencing negative affect is emotion regulation (ER). Using this framework, a possible explanation for the correlation seen between the *GABRB3* risk allele and increased eating expectancies in our study cohort could be that individuals with dysregulated GABA_A_ signaling are more likely to use food to regulate their affective states and count on food to improve their mood. Consequently, these individuals may be at greater risk of jeopardizing their maintenance of weight loss after surgery by continuing to rely on food to cope with negative mood states.

The *GABRB3* gene was also significantly correlated with scores on the Food Cravings Questionnaire (FCQ), showing a similar outcome as previously discussed with the EEI. Patients in our study with two copies of the risk allele had scores indicative of greater food cravings compared to patients with one or no copies of the risk allele. Items on the FCQ measure the individuals’ perceived ability to control their cravings and the extent to which food occupies their thoughts and influences their emotions. The correlation from this questionnaire adds further support to the hypothesis that homozygous carriers of the *GABRB3* risk allele are at an increased risk of using food to alleviate negative mood states and because of this coping strategy, may over-eat in a manner that leads to weight regain after bariatric surgery.

A negative correlation is presented between the *COMT* gene and scores on the PSQI. The inverse relationship we saw indicates that homozygous carriers of the *COMT* risk allele reported lower scores on the PSQI, associated with good sleep quality, compared to patients with one or no risk alleles. There is some preexisting research on the association between *COMT* polymorphisms and sleep; however, not all studies come to the same conclusion. Investigation into this relationship is based on evidence that individuals homozygous for the Val allele show higher *COMT* activity and lower dopaminergic signaling in the prefrontal cortex than subjects homozygous for the Met allele. Due to the crucial role of *COMT* in metabolizing DA, it has been suggested that the *COMT* gene polymorphism impacts cognitive function, sleep–wake regulation, and sleep pathologies [55]. On the other hand, the current literature has contributed investigations that lack support for the impact of the *COMT* gene polymorphism on arousal regulation and sleep [56]. Given the high rates of sleep apnea among obese populations, examining the possible influence of the Val158Met polymorphism on sleep quality is of relevance and should be further examined.

Additionally, our results revealed a significant correlation between the *OPRM1* gene and scores of the DERS. As previously discussed, emotion regulation is a major factor influencing outcomes after surgery for patients. The A118G polymorphism of the *OPRM1* gene has been associated with lower mRNA transcription and translation and a dampening of opioid signalling efficiency in the brain [57]. Carriers of the G allele resulting from this gene polymorphism have been shown to display increased emotional dysregulation and frequency of mood disturbances and have been shown to warrant potential as a modulator of sensitivity to stressors [57,58,59]. These findings help lend an explanation and support to the significant correlations discussed here. No patients in our study were homozygous carriers of the G risk allele; however, heterozygous carriers reported higher scores on the DERS, indicating that they experience greater than normal difficulties in regulating their emotions.

Variations in the genes associated with these neurotransmitters can lead to insensitivity to positive reinforcers. Notably, maternal ingestive behavior can alter the expression of mesocorticolimbic genes implicated in the regulation of dopamine and opioid function in offspring. This finding was reported from a study investigating receptor binding differences in mice prenatally exposed to low protein (LP) or high fat (HF) diets. Results from this experiment showed that LP but not HF offspring had significantly elevated mu-opioid receptor (MOR) and D1R binding in the brain, highlighting the possible outcomes of suboptimal gestational diets and how they may increase risk for obesity, addiction, and ADHD [60].

Self-report outcomes on the DERS were also significantly correlated with pre-surgery and at six months post-surgery BMI. The positive correlations for both study time points suggest that having greater emotional regulation difficulties is associated with a greater BMI. This relationship has been thoroughly examined in previous studies, providing a depth of evidence for the significant correlation of obesity and emotional dysregulation [61,62,63]. Individual personality and temperament analyzed within the DERS relates to long-term outcomes after bariatric surgery. In fact, certain dispositional tendencies are shown to be less optimal for patients after surgery, highlighting the usefulness of a psychosocial analysis prior to surgery to inform a targeted approach to improving outcomes [64].

While heterosis is an important concept in genetics as evidenced by many studies including the lesson learned with the *DRD2* A1 allele and RDS [12,36], because of high allelic presence of the reward genes as measured in GARS in the general population, we chose to utilize a very conservative approach whereas zero vs. two copies provided highly accurate statistical power. However, our significant finding with the *DRD4* polymorphism further supports a potential heterosis phenomenon.

### Limitations

Limitations to this study include a small sample size because of the COVID-19 pandemic. Additionally, the genetic and psychosocial measures of these patients can be considered cofactors. Other epigenetic factors were not considered in this study. These subjects will be monitored for bodyweight data as they relate to genetic and psychosocial data to determine long term outcomes in addition to the present study. It is important to highlight the fact that the GARS test utilized in this study has not been independently validated by scientists across the globe. However, while this is true, it is our contention that currently there are 74 articles listed in Pubmed as of 21 June 2023. The development of this genetic assessment was initiated by Blum’s laboratory in 2014 published in *Molecular Neurobiology* [33]. Since that time the same authors have published human trials concerning various applications of GARS in clinical medicine with significantly significant results [11,13,14,16,65,66,67,68,69,70,71,72,73,74,75,76,77,78]. Moreover, since 1990, when our laboratory published the association of the *DRD2* Taq A1 allele and severe alcoholism in *JAMA*, there has been an explosion of genetic candidate association studies, including GWAS. In order to statistically validate the selection of the risk alleles analyzed by GARS, Blum’s laboratory applied a strict analysis to studies that investigated the association of each polymorphism with AUD or AUD-related conditions published from 1990 until 2021. The total number of patients evaluated was 74,566. This evaluation calculated the Hardy–Weinberg Equilibrium of each polymorphism in cases and controls. If available, the Pearson’s *χ*^2^ test or Fisher’s exact test was applied to comparisons of the gender, genotype, and allele distribution. The statistical analyses found the OR, 95% CI for OR, and a post-risk for 8% estimation of the population’s alcoholism prevalence revealed a significant detection. The OR results showed significance for *DRD2*, *DRD3*, *DRD4*, *DAT1*, *COMT*, *OPRM1*, and *5HTT* at 5% [68]. While most of the research related to GARS is derived from our laboratory, we are encouraging more independent research to confirm our findings. In addition, large GWAS from Yale recently revealed that for example, in 1.2 million veterans, the top candidate gene to associate with depression was the *DRD2* gene polymorphism tested in GARS [79] was at 1 × 10^−7^. In addition, using the same 1.2 million veterans, scientists from Duke performed GWAS and found that the top gene was also the *DRD2 AT* 10^−12^ [80]. Additionally, scientists from Washington University also found that the *DRD2* Taq A1 polymorphism in GWAS in over 1 million subjects also identified this loci underlying multiple substance use disorders. In sum, we are very encouraged by these first ever results linking known reward gene risk polymorphisms to a number of metrics related to clinical outcomes post bariatric surgery. Knowledge of precise polymorphic associations can help in the attenuation of: guilt and denial; corroboration of family gene-o-grams; assistance in risk-severity-based decisions about appropriate therapies, including pain medications and risk for addiction; choice of the appropriate level of care placement (i.e., inpatient, outpatient, intensive outpatient, residential); determination of the length of stay in treatment; determination of genetic severity-based relapse and recovery liability and vulnerability; determination of pharmacogenetic medical monitoring for better clinical outcomes (e.g., the A1 allele of the *DRD2* gene reduces the binding to opioid delta receptors in the brain, thus, reducing Naltrexone’s clinical effectiveness); and supporting medical necessity for insurance scrutiny. Of course, based on our results herein, we encourage further research to confirm and extend these seemingly very important results. Our futuristic perspective is if these results are subsequently confirmed in much larger studies, we encourage global implementation.

## 5. Conclusions

The mean change in BMI (10.3 kg/m^2^ ± 4.3) observed was similar to the six-month weight loss post bariatric surgery reported in the literature. By utilizing genetic testing, we were able to construct a genetic profile for each patient that would inform health care providers and patients regarding individual risk for alcohol and substance abuse. Beyond calculating total risk scores, risk alleles from each gene, we assessed associations with other factors such as patients’ change in BMI, % excess weight loss, and psychosocial questionnaire scores. This risk may very well predict other hazardous post-surgery RDS behaviors, such as smoking and gambling. The subsequent information may be then used clinically to counsel patients and potentially provide additional adjunct support and care. The multi-disciplinary approach used here considers how RDS interacts with factors that influence maintenance of weight loss after surgery and overall quality of life. The use of genetic testing in bariatric surgery populations as a predictive measure for risk outcomes is a novel approach which can be incorporated into personalized medicine. This initial study is the first to combine a psychosocial questionnaire and genetic testing as an approach towards identifying negative outcomes after surgery. As a predictive tool, genetic testing continues to show meaningful utility in the ongoing goal of fine-tuning precision medicine and enhancing treatment strategies for patients.

## Figures and Tables

**Figure 1 jpm-13-01164-f001:**
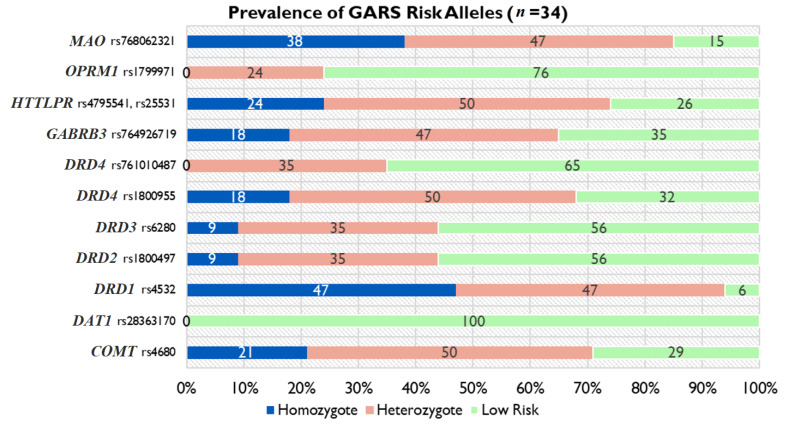
GARS results represented as the percent frequency of gene type for each risk allele analyzed within the patient population (*n* = 34). The homozygote category represents having two copies of the risk allele, heterozygous represents one copy of the risk allele, and the low-risk category represents no copies of the risk allele per gene.

**Figure 2 jpm-13-01164-f002:**
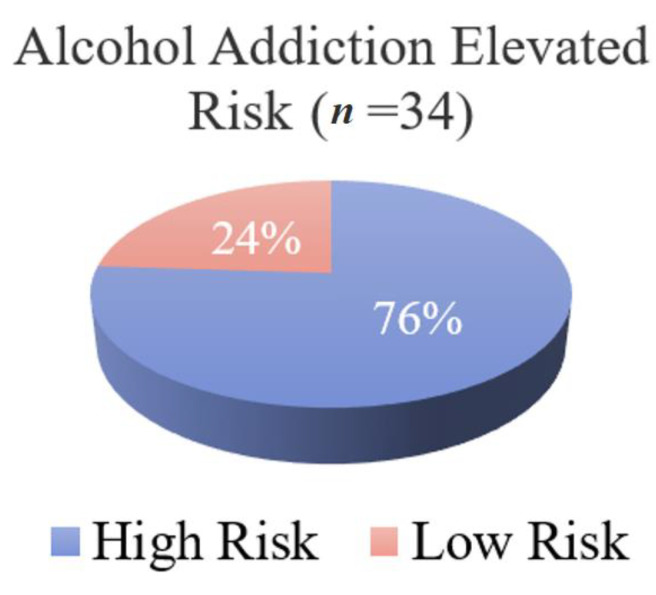
Frequency of patients with a GARS score qualifying them for high risk or low risk of alcohol addiction. Risk category is based on total number of GARS risk alleles.

**Figure 3 jpm-13-01164-f003:**
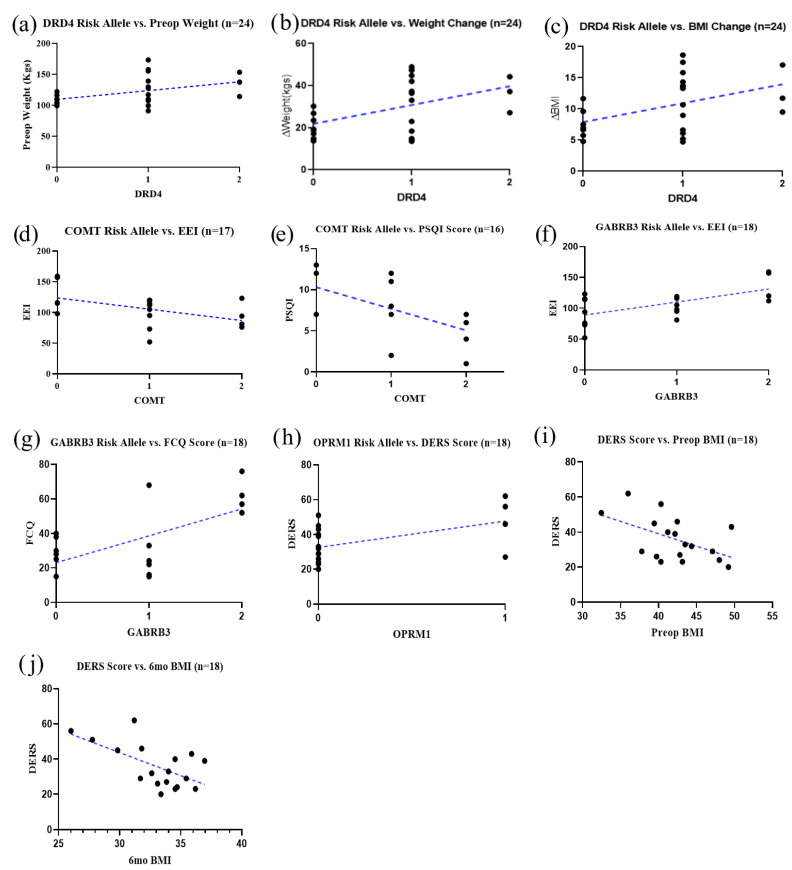
(**a**–**j**). BMI and specific psychosocial inventory outcomes are correlated with GARS: (**a**) *DRD4* risk allele positively correlated with preop weight (*r* = 0.4331, R^2^ = 0.1876, *p* = 0.0345)*;* (**b**) *DRD4* risk allele positively correlated with change in weight (*r* = 0.4726, R^2^ = 0.2234, *p* = 0.0197); (**c**) *DRD4* risk allele positively correlated with change in BMI (*r* = 0.4577, R^2^ = 0.2095, *p* = 0.0245) (**d**) *COMT* risk allele negatively correlated with EEI scores (*r* = −0.4983, R^2^ = 0.2483, *p* = 0.0418); (**e**) *COMT* risk allele negatively correlated with PSQI scores (*r* = −0.5482, R^2^ = 0.3005, *p* = 0.0279); (**f**) *GABRB3* risk allele positively correlated with EEI scores (*r* = 0.6161, R^2^ = 0.3796, *p* = 0.0084); (**g**) *GABRB3* risk allele positively correlated with FCQ scores (*r* = 0.6373, R^2^ = 0.4062, *p* = 0.0044); (**h**) *OPRM1* risk allele positively correlated with DERS scores (*r* = 0.5228, R^2^ = 0.2733, *p* = 0.0260). (**i**) DERS scores negatively correlated with preop BMI (*r* = −0.5142, R^2^ = 0.2644, *p* = 0.0290); (**j**) DERS scores negatively correlated with BMI 6 months post-surgery (*r* = −0.6137, R^2^ = 0.3766, *p* = 0.0068).

**Figure 4 jpm-13-01164-f004:**
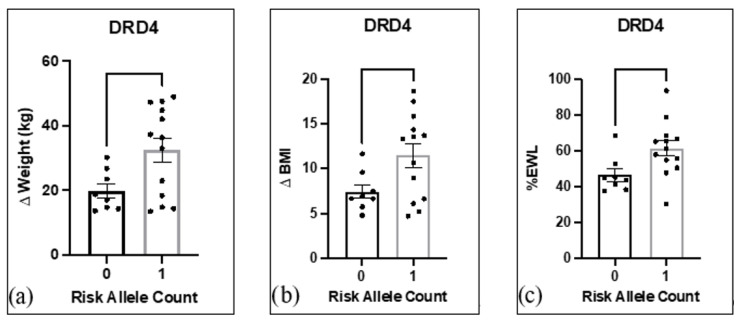
(**a**–**c**). Heterosis in *DRD4* gene: (**a**) significant difference (*t* = 2.400, *p* = 0.03) of mean change in weight at 6 months between patients with 0 and 1 copies of the *DRD4* risk allele; (**b**) significant difference (*t* = 2.234, *p* = 0.04) of mean change in BMI at 6 months between patients with 0 and 1 copies of the *DRD4* risk allele; (**c**) significant difference (*t* = 2.418, *p* = 0.03) of %EWL at 6 months between patients with 0 and 1 copies of the *DRD4* risk allele. Data are M ± SEM Student’s *t* test.

**Table 1 jpm-13-01164-t001:** Self-Report Data (Demographics, weight, BMI) (*n* = 26).

**Age (years)**	M = 47 ± 12, range: 22–72
**Sex**	90% Female
**Race**	85% White
**Weight (kgs)**	M = 118, SD = 20.8
**BMI**	M = 43, SD = 6.0
**Surgery**	26% RYGB (bypass) 74% Vertical Sleeve Gastrectomy
**Childhood Weight Status**	8% Underweight; 38% Healthy Weight; 42% Overweight 12% Obese
**Marital Status**	27% Single/Never Married; 41% Married/Living with Spouse; 11% Living with Intimate Partner 14% Divorced 7% Widowed
**Employment**	51% Full-time; 11% Part-time 11% Not employed, but looking 27% Not employed, not looking
**Education**	96% Graduated from High School/GED 50% College Degree (Associates/Bachelors) 15% College Degree (Masters)
**Income Last Year**	27% Less Than $10,000 27% $1500–$55,000 23% $55,000–$99,000 12% $100,000 and over

Mean patient weight data as well as demographic data from patients based on self-report measures reported preoperatively and Electronic Health Records from the bariatric clinic (*n* = 26).

**Table 2 jpm-13-01164-t002:** BMI/Weight Loss Outcomes at 6 Months Post-Surgery (*n* = 24).

	Total Population (*n* = 24)	VSG (*n* = 19)	RYGB (*n* = 5)
**Mean BMI ± SD**	33.3 ± 3.7	32.7 ± 3.6	35.5 ± 4.2
**Mean** **△** **BMI ± SD**	10.3 ± 4.3	9.8 ± 4.3	12.1 ± 4.4
**%EWL ± SD**	56.0 ± 13.8	56.3 ± 14.3	54.5 ± 14.7

Summarized BMI, △BMI from preop to 6 months post-op, and percent excess weight loss (%EWL) results from patients of the total study population (*n* = 24) and those who underwent either VSG (*n* = 19) or RYGB surgery (*n* = 5).

**Table 3 jpm-13-01164-t003:** Psychosocial Questionnaire Results. Mean (SD).

Eating Attitudes Test-26	Total: 14.9 (8.1)
Food Cravings Questionnaire—Trait Reduced (FCQ-T)	- Domain Control: 2.3 (1.17) - Thoughts: 2.1 (1.23) - Plans: 2.5 (1.57) - Emotions: 2.4 (1.33) - Cues: 2.7 (1.54)
Eating Expectancies Inventory	- Manage Negative Affect: 2.91 (2.02) - Pleasurable and Useful as a Reward: 3.62 (2.23) - Feeling Out of Control: 3.12 (2.11) - Enhances Cognitive Competence: 2.69 (1.82) - Alleviates Boredom: 3.35 (2.23)
Modified Yale Food Addiction Scale 2.0	Mean Symptom Count (SD): 1.32 (1.23) No Food Addiction (%): 61 Mild (%): 31 Moderate (%): 4 Severe (%): 4
Weight-Influenced Self Esteem Questionnaire	M (SD): 1.6 (1.3)
Difficulties in Emotion Regulation Scale—Short Form	Total Mean (SD): 33.81 (10.96) - Total w/o Awareness: 27.5 (10.52) - Awareness: 6.35 (2.46) - Clarity: 4.61 (1.80) - Goals: 7.58 (3.88) - Impulse: 4.23 (2.3) - Non-acceptance: 5.65 (2.67) - Strategies: 5.38 (2.89)
Center for Epidemiological Studies Depression Scale	Total Score (Mean, range): 12.7, 0–35 No Depression (%): 69 Mild Depression (%): 8 Probable Depression (%): 23
Chronic Stress Index	Perceived Everyday Unfair Treatment (Mean Score): 1.8 Major Negative Life Events in Past Year: 1.13
Quality of Life Enjoyment and Satisfaction Questionnaire	M (SD): 3.24 (0.89)
Pittsburgh Sleep Quality Index	M (SD): 8.0 (3.74)

Summary of scored outcomes from self-report psychosocial questionnaires completed by patients prior to surgery (*n* = 26). Mean score totals and subscale scores for each inventory.

## Data Availability

Data is available from the corresponding author. Blum is the inventor of GARS and has assigned patent rights to Igene LLC, and Transplicegen Holdings LLC.

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
