# Peer review of "The First Exploratory Personalized Medicine Approach to Improve Bariatric Surgery Outcomes Utilizing Psychosocial and Genetic Risk Assessments: Encouraging Clinical Research"

_jpm, 2023, doi:10.3390/jpm13071164_

Round 1

Reviewer 1 Report

Dear authors, 

The field of research you are investigating is very novel and interesting. I find your paper well written. 

However, in my opinion, the data you provide is not robust enough to make conclusions. I do not agree this paper describes encouraging global implementation.

Firstly, the titel suggest a global (?) implementation for an intervention that has not yet been validated. Moreover this type of genetic testing has not proven any beneficial effect on patients. I find this misleading. 

My second concern is the statistical methods used, i did not find corrections for multiple testing. There was no power calculation. The study population is very small. I believe this paper is missing a proper discussion section (strengths and weaknesses;  what is the meaning of the findings summed up in the discussion; etc.)

Reviewer 2 Report

I would like to congratulate the authors for their interesting manuscript

My comments are for minor revision: 

1. There are repeated words in lines  75 and 273

2. It is important to mention that obesity and weight regain have multifactorial etiology such as gut hormone balance changes, postprandial hypoglycemia, etc. I suggest to address this in your discussion, since genetics and behavior are cofactors.

3. Please add a limitation section- the N is relatively small, the short-term outcomes (it will be interesting to see long term outcomes) as it also affects the significance of conclusions 

Author Response

My comments are for minor revision: 

  1. There are repeated words in lines 75 and 273:

Reply: The repeated word in line 75 was corrected but the one in line 273 was not identified.

  1. It is important to mention that obesity and weight regain have multifactorial etiology such as gut hormone balance changes, postprandial hypoglycemia, etc. I suggest to address this in your discussion, since genetics and behavior are cofactors.

Reply: This has been added to the limitations section of the discussion. See line 415 under  newly added Limitations section.

  1. Please add a limitation section- the N is relatively small, the short-term outcomes (it will be interesting to see long term outcomes) as it also affects the significance of conclusions.

Reply:  This has been expanded on in line 108 of the methods and in the newly added Limitations section (Page 13).

Reviewer 3 Report

congratulate the authors for seeking additional avenues to continuing to improve all multidisciplinary aspects of the postoperative bariatric care so maximize results. Psychologicla support should not stop postop, and identifying patients at at a hightened risk for addictive or reward seeking behavior might prove beneficial in so that these patients should maintain a closer psychological follow up to prevent weight recidivism (or development of other addictive behaviors). a larger sample size would have been nicer, though the pandemic limitations were disclosed. 

Author Response

congratulate the authors for seeking additional avenues to continuing to improve all multidisciplinary aspects of the postoperative bariatric care so maximize results. Psychologicla support should not stop postop, and identifying patients at at a hightened risk for addictive or reward seeking behavior might prove beneficial in so that these patients should maintain a closer psychological follow up to prevent weight recidivism (or development of other addictive behaviors). a larger sample size would have been nicer, though the pandemic limitations were disclosed. 

Reply: Small sample size has been expanded on in line 109 of the methods and in the newly added “Limitations” section. Additionally, long-term psychological data will be collected and assessed in future studies.

Round 2

Reviewer 1 Report

Dear authors,

I appreciate the change made to the title, as well as the first paragraph of the limitation section (413-419). 

In my opinion the rest of the limitation section is more suitable for the introduction (412-455). To me the number of self citations of this section is unnecessary and I still miss discussion of limitations. 

I was confused when reading section: 446-455. Afterwards I noticed that this is +/- exact copy of abstract of https://doi.org/10.29014/IJGD-115.000015; which is not fully applicable for this article. 

In conclusion: to me this is an exploratory study. I remain reluctant to accept this paper for publication. I believe it is up to the editor to decide if the statistical methods used and the way the data is presented meet the quality criteria of this journal. 
